# Trend of Correlations between Psychological Symptoms and Socioeconomic Inequalities among Italian Adolescents: Cross-Sectional Study from 2006 to 2018 in Tuscany Region

**DOI:** 10.3390/ijerph20156473

**Published:** 2023-07-28

**Authors:** Dario Lipari, Bianca Maria Bocci, Cesare Rivieri, Elena Frongillo, Antonella Miserendino, Andrea Pammolli, Claudia Maria Trombetta, Ilaria Manini, Rita Simi, Giacomo Lazzeri

**Affiliations:** 1Post Graduate School of Public Health, University of Siena, 53100 Siena, Italy; 2Department of Molecular and Developmental Medicine, University of Siena, 53100 Siena, Italy; pammolli2@unisi.it (A.P.);; 3Research Center on Health Prevention and Promotion (CREPS), University of Siena, 53100 Siena, Italy; 4Interuniversity Research Centre on Influenza and Other Transmissible Infections (CIRI-IT), 16132 Genoa, Italy

**Keywords:** HBSC study, adolescence, psychological symptoms, trends, socio-economic factors, mental wellbeing, gender, family affluence scale

## Abstract

Adolescence is a critical moment in life; people become individuals, create new relationships, develop social skills and learn behaviours that they will use for the rest of their lives. During this phase, adolescents establish patterns of behaviour that can protect their health. This study aims to 1. assess the presence of psychological disorders in adolescents of both genders, 2. determine their relation to socio-economic differences based on the Family Affluence Scale (FAS), and 3. assess trends from 2006 to 2018. Data were collected from the Italian Health Behaviour in School-aged Children survey given to a representative sample of Tuscan adolescents aged 11–15 years. Participants (*n*. 12,550) filled out questionnaires to assess whether psychological symptoms such as feeling low, irritability, nervousness, or sleeping difficulties manifested weekly or more often over the past six months. For the study we utilized a cross-sectional survey method and linear regression to examine the association between psychological symptoms (dependent variable measured on an interval scale (0–16)), gender and FAS. We conducted similar analyses using logistic regressions for each of the four symptoms. An increase in psychological symptoms in both genders was revealed between 2006 and 2018, with a statistically significant presence in females: 0.29 (95% Confidence Interval [CI], 0.17 to 0.41), 1.43 (95% CI, 1.37 to 1.48) and 1.43 (95% CI, 1.34 to 1.52) in low-, medium- and high-affluence families, respectively; whereas males presented 0.14 (95% CI, 0.01 to 0.27), 0.71 (95% CI, 0.65 to 0.77) and 0.31 (95% CI, 0.22 to 0.39), respectively. The probability of a predictive episode of psychological symptoms (feeling low, irritability, nervousness, sleeping difficulties) occurring weekly, or more, often was greatly increased in females of all socioeconomic classes. These findings suggest that the increase in psychological disorders in adolescents should be considered a public health problem and further investigated through longitudinal studies and continuous monitoring of health trends.

## 1. Introduction

Adolescence is a critical and complex stage of development in everyone’s life. It is a period of transition that goes from puberty, the phase in which physical and sexual development begins, until the end of adolescence, usually around 19–20 years of age. According to the World Health Organization (WHO), adolescence occurs in the period of life ranging from 10 to 19 years of age [1]. During adolescence we lay the foundations for conscious, autonomous, and healthy growth. Discomfort caused by physical changes and cognitive development is experienced, this is the phase in which their alignment with body identity and personal sexual identification is formed, and psychosocial awareness grows. Often these events occur suddenly before adolescents have acquired the psychological skills to be able to effectively manage them. These difficulties inevitably affect adolescents’ perception of themselves as well as others, how they think and the decisions they make relating to the world around them [2,3].

During this phase, young people acquire patterns of behaviour, such as those related to physical activity or nutrition. They may be driven to experiment with new experiences, such as the use of psychotropic substances, which could imply the development of potentially negative effects on their physical and mental health and greatly increase the risk of developing addictions, emotional disorders, and behavioural problems. A further critical issue is represented by the first approaches to the sphere of sexuality. In fact, this is the period in which young people begin to know and explore their bodies and approach others in a more adult way. Lack of knowledge and experience can lead to risky sexual behaviours, such as unprotected relationships, promiscuity and unsafe sexual partners. All this tends to increase the risk of contracting sexually transmitted infections (STIs) or unwanted pregnancies [4]. Other issues frequently associated with this stage are those related to mental distress such as anxiety, eating disorders and depression.

These disorders can be caused by several promoting factors such as social isolation, having to deal with unrealistic expectations or with difficulty in understanding and managing their own emotions. Sometimes, manifestations of the psychological disorders exposed above can assume the characteristics of actions aimed at escaping from personal or family problems, which could affect their own health and that of others, even in their future [1]. Several authors have shown how so-called “adolescent upheavals” are, first and foremost, a cultural product, thus introducing a radical shift in perspective. The way in which adolescent transition manifests itself is affected by family influence and cultural and social context [3,5].

There is a big debate about the definition of social capital and how it can be measured; the two main elements used to identify social capital are relationships and resources. By social relations, individuals or groups of people can gain access to social resources. In our study context, social capital is defined as the resources derived from a network of relationships that people can access and mobilise to improve their lives [6].

Social capital has typically been defined as an asset that improves the resilience and capacities of people and communities to sustain and support health and well-being by protecting against poor health conditions, providing social support and making collective action easier [7,8]. Regarding children and adolescents, data suggest that social capital and social support in the family and school context may act as protective factors for their well-being [9,10].

Socioeconomic inequalities have a heavy impact on people’s mental health and well-being. Lack of access to services, exposure to social problems and inadequate family support can contribute to the onset of psychological problems among adolescents. Since it is not easy to change the income factor, it is important to work on what can be changed. In this regard, it is useful to provide young people with a safe environment in which they can express their feelings and concerns and provide a beneficial social support system that can mitigate the onset of these evident types of psychological distress and health problems [11].

The necessity to investigate adolescents’ psychophysical development arises from the awareness of the risks they face in this delicate phase of life. With this aim, the surveillance and research project “Health Behaviour in School-aged Children” (HBSC) was born [12]; it is an international study that precisely focuses on investigating adolescents’ health and lifestyle habits. Thanks to this, the aims of our study were to investigate and assess the presence of psychological disorders, such as feeling low, irritability, nervousness, and difficulty sleeping, in Tuscan adolescents of both genders and to test the correlations between the type of disorder and socioeconomic differences (through the Family Affluence Scale (FAS)). Furthermore, the trend in these disorders from 2006 to 2018 was analysed.

## 2. Materials and Methods

### 2.1. Study Design and Population

The data presented in this study were taken from HBSC surveys conducted in Tuscany over four years (2006, 2010, 2014, 2018). The methods used to gather the data are described in detail below [13,14]. Passive parental consent was necessary to participate in the survey. The Ethics Committee of the National Institute of Health, which approved the protocol, established that an adolescent would be included by default unless his or her parents chose to “opt-out” by explicitly refusing consent. Data collection was anonymous, and the demographic information collected (gender, year and month of birth, nationality, nationality of parents) does not permit identification of the individual adolescent. HBSC data is randomly collected from a sample of schools (also known as cluster sampling) every fourth year for adolescents aged 11 and 13, who are in the first and last years of middle school, and 15, who are in the second year of high school. The HBSC study uses an anonymous, self-administered questionnaire which complies with international standards and is distributed in schools. Many professionals were involved in the HBSC national network and contributed to its activities. Head teachers, teachers, and other school staff, as well as health workers, were involved; the specific expertise and work done by each of these professionals can be found elsewhere [13]. Standardised scientific methods were used to collect the data ensuring reliable and valid results; the questionnaire items used were consistent across the four survey waves.

### 2.2. Sample

The same protocols, in terms of target groups, sampling and data collection, were used in all surveys. Systematic cluster sampling (schools), stratified by administrative district, was applied in each of the four waves from which the samples were drawn [13,14].

To obtain representative samples for each age group, a stratified cluster sampling method was used, with school classes as the primary sampling unit. The sampling methodology and data collection procedures of the adolescent population in Tuscany were analysed in such a way as to improve the generalizability of the study results. Data on sampling methodology and survey data collection can be found elsewhere [13].

We included all participants from the 2006 (*n* = 3564), 2010 (*n* = 3291), 2014 (*n* = 2511) and 2018 (*n* = 3184) surveys, resulting in a total sample of 12,550 participants.

The HBSC questionnaire included an 8-item symptom checklist which asked students to report the frequency (once a week or more often, once a month, or rarely/never) of physical and psychological symptoms experienced in the previous 6 months. To measure psychological symptoms, we used the 4-item psychological subscale checklist, a tested and reliable tool for assessing the psychological health of adolescents. The subscale items focus on psychological or emotion-related problems, including feeling low, being irritable or in a bad mood, feeling nervous and having difficulty sleeping. Students rated the frequency of each symptom on a 5-point scale from 0 (rarely or never) to 4 (about every day). This has been validated on samples of adolescents as a measure of psychological health and shows good internal consistency (Cronbach’s α = 0.75) and convergent validity with indicators of emotional problems [15].

The HBSC FAS was used to measure socioeconomic position. The FAS is an index that requires 4 common indicators of family wealth (number of cars, computers, whether the child has his or her own bedroom, and how often the family travels for vacations). This scale is valid and less susceptible to bias than what adolescents report regarding their family’s income, occupation or parent’s level of education. For each survey wave, the FAS was classified into approximate tertiles based on proportional ranks within age and gender groups (low, medium, high) [15,16].

### 2.3. Statistical Analysis

For the study we used a cross-sectional survey method. We utilized a linear regression to examine the association between psychological symptoms (dependent variable measured on an interval scale (0–16)) and gender and FAS (as a factor of independent variables), adjusted for year of survey and age of adolescent (as covariate independent variables). We included all 2-way and 3-way interaction terms between gender, family affluence and year of survey.

We conducted similar analyses using logistic regressions for each of the 4 symptoms (feeling low, feeling irritable, feeling nervous, difficulty sleeping), modelling the log odds of experiencing each symptom once a week, or more, compared to once a month or rarely/never (so, in this case, psychological symptoms were taken as dummy variable). We estimated the predicted scores (from the linear models) and predicted probabilities (from the logistic models) for each gender-affluence category by survey year and calculated the 12-year differences in symptom score and probability from the regression output. Statistical analyses were conducted in SPSS v.26 (IBM Corp, Armonk, NY, USA).

## 3. Results

The distribution of family affluence groups was similar between examined groups during the different survey waves (Table 1).

Scores derived from the analysis of surveys reporting psychological disorders perceived once a week, or more, in the previous 6 months increase over the years among girls in all three affluence groups (low-, middle-, high-) by 0.29 (95% confidence interval [CI], 0.17 to 0.41), 1.43 (95% CI, 1.37 to 1.48) and 1.43 (95% CI, 1.34 to 1.52) points, respectively. Among boys, psychological symptom scores also increased over the same period, although less so than girls, by 0.14 (95% CI, 0.71 to 0.19), 0.71 (95% CI, 0.65 to 0.77) and 0.31 (95% CI, 0.22 to 0.39) (Table 2). There was no evidence of an interaction between gender and affluence in the model (*p* > 0.05), suggesting that each associating factor trends independently from one another (Table 3).

As illustrated in Figure 1, trends in psychological symptoms varied by specific symptoms, gender and affluence. The probability of feeling depressed and irritable once a week, or more often, decreased in the low affluence category while the same symptoms increased for the girls of medium affluence, whereas feeling nervous and difficulties sleeping were shown to increase across all affluence groups.

Conversely the probability of feeling low and nervous once a week or more decreased for the low affluence group while symptoms of feeling low and irritable increased for medium affluence boys, and the probability of feeling nervous increased among the low and medium affluence groups.

The probability of feeling low decreased while feeling irritable increased among boys from the high-affluence group (Table 4).

Psychological symptoms were predicted by a linear regression model, adjusted for gender (boys/girls), family affluence scale, survey year and adolescent age, and for all two- and three-way interaction terms between gender, family, and survey year, and weighted using survey weights.

## 4. Discussion

The aim of this study was to investigate the association between psychological disorders and socioeconomic status in adolescents in Tuscan schools, aged 11, 13 and 15 years. According to the WHO, “Mental health is a state of mental well-being that enables people to cope with the stresses of life, realize their abilities, learn well and work well, and contribute to their community” [17,18]. It is thus clear that mental health is more than the absence of mental disorders. The link between socioeconomic factors and health is so close that the WHO has a dedicated Commission that deals with it and investigates population outcomes (Commission on Social Determinants of Health). In 2012, the World Health Assembly passed resolution 65.8, which endorsed the Rio Political Declaration on Social Determinants of Health and emphasized the need for “delivering equitable economic growth through resolute action on social determinants of health across all sectors and at all levels” [19].

If the economic aspect has such huge impact at the social level, we can infer that its effects are also felt at the adolescent level, the latter being individuals who do not yet have an established emotional behavioural attitude [20,21,22,23].

In this context, it is important to consider social capital. We can distinguish the social capital into a horizontal component, which generally reflects ties between individuals or peer or near-peer groups, and a vertical component, which describes ties between hierarchical or unequal individuals or groups who have different access to resources and power [24]. Applied to a school context, horizontal social capital refers to the ties between classmates, whereas vertical ones refer to connections between children and teachers [25,26]. After examining how the two genders perform regarding psychological symptoms, we can see that girls have higher overall scores than boys in all three income brackets, and this gender gap tends to remain stable over time.

This result tends to corroborate a growing strand of research that emphasizes that there is a trend of increased emotional, psychological, and somatic upsets in girls [27].

As reported by other authors who have investigated the mental health status of Italian adolescents from 2010 to 2018, it can be seen that our data is perfectly aligned with national data, but also with that of other European countries [27,28,29,30,31,32]. Authors have taken note that girls, compared to boys, seem to be affected sooner by socio psychological symptoms and they are more pronounced. Possible explanations, beyond those exposed before, could be related to the tendency that girls experience more internal rather than external symptoms, but also are exposed to more restricted gender rules, experience higher levels of body dissatisfaction, and have a higher perception of stress related to school performance [28,33,34,35]. In adolescents, the emotional components of well-being tend to be more liable to fluctuation than life satisfaction, which is usually described as a more stable component. However, these findings underscore the need to consider mental well-being as a multidimensional construct and suggest the need to further understand associations between risk factors and different aspects of mental well-being [27]. A major environmental factor involved in well-being is social support [28,33,36], both from family and peers, which can also be a crucial protective factor during economic downturns [28,29,30,31,32,33,34,35,36,37]. According to literature, the family setting could influence well-being not just directly influencing individual relationships but also as a safe place to learn to handle the school environment and adversity [38,39]. Puntscher et al. found that family and friendship ties were positively correlated with happiness [40]. It is therefore likely that adolescents rated their life satisfaction according to their economic conditions, whereas they rated their happiness according to their psychosocial conditions and thus social capital [41]. The positive relationship between social capital and happiness may also be attributed to the psychological benefits, such as self-esteem, that result from a sense of ownership and control [42,43].

As revealed by other studies, social capital plays a protective role in the presence of FAS inequalities [40,44,45,46].

Adolescents from poor families but with a high level of social capital are more likely to be protected from well-being disparities than adolescents from poor families who also have a low level of social capital. A sense of family ownership, support for family autonomy and family control mediate the effects of FAS on mental well-being [41].

Studies concerning the mediating role of social capital suggest that the family context (home) is a stronger protector of adolescents’ well-being than other social contexts (school) [41]. Thus, only a sense of family belonging, support for family autonomy and family control were found to be protective towards mental wellbeing.

Therefore, having evaluated the trend and considering its general concordance with the national and international data examined, and considering the sociological point of view, we believe that a multidisciplinary and state-specific analysis would be useful; in this way we will be able to assess, with a greater amount of data, how the variability found is conditioned by the factor “family” (in certain family contexts it helps adolescents not to be affected by the socioeconomic gap, in others it could be a negative protective factor).

This study has several strengths, such as the large, regionally representative sample of Tuscan adolescents and a standardized international protocol for data collection.

Limitations include the cross-sectional and self-reported nature of the data. In cross-sectional studies the temporal link between outcome and exposure cannot be determined because both are examined simultaneously; another limitation of the study is that, for some of its aspects, it is not possible to analyse all the variables involved in the problem.

## 5. Conclusions

Children and adolescents, especially the most vulnerable, as already highlighted in the UN report at the beginning of the pandemic, have been heavily damaged, not only physically but also psychologically and emotionally by the effects of the pandemic. Adolescents have rarely been the subject of specific attention and interventions [47]. The social rules they are asked to follow stand in contradiction to the natural impulses existing during this phase of the life cycle in which the person is strongly involved in exploring the world, in search of autonomy and new experiences, in building meaningful relationships outside their family of origin, in attaching importance to values such as opening up to change, in exploring future plans and, not least, in building a renewed awareness of their own bodily identity [48].

All the symptoms we investigated in this study were found to be an alarm as to the effects of social isolation on mental health.

These data suggest that the increase in psychological disorders in adolescents should be considered a public health problem. It is necessary to address the problem to better understand the determinants of their psychological well-being and it should be further investigated through longitudinal studies and continuous monitoring of health trends. In addition, more emphasis should be placed on primary prevention and promotion of adolescent well-being, considering age and gender differences.

## Figures and Tables

**Figure 1 ijerph-20-06473-f001:**
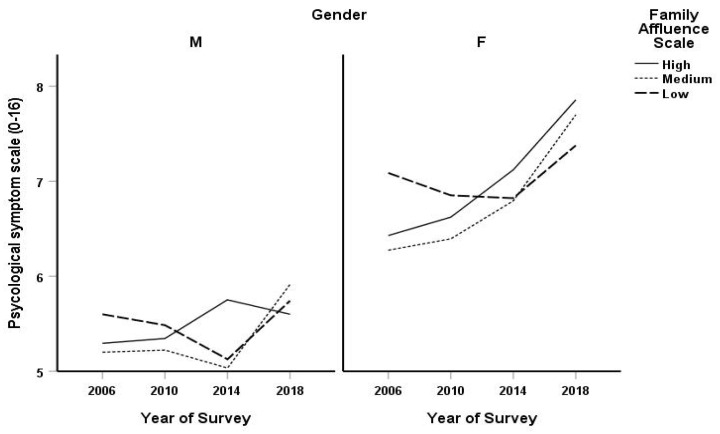
Time trend of psychological symptoms by gender and family affluence scale among Tuscan adolescents 2006–2018.

**Table 1 ijerph-20-06473-t001:** Distribution of Family Affluence Scale by Year and Gender in the Tuscany HBSC 2006–2018 Surveys.

		2006, *n* (%)	2010, *n* (%)	2014, *n* (%)	2018, *n* (%)
Girls					
Family affluence scale	Low	305 (16.9)	128 (8.1)	230 (17.5)	231 (14.6)
	Medium	1092 (60.5)	869 (54.7)	785 (59.7)	989 (62.5)
	High	408 (22.6)	592 (37.3)	300 (22.8)	362 (22.9)
Boys					
Family affluence scale	Low	239 (13.6)	134 (7.9)	206 (17.2)	243 (15.2)
	Medium	1024 (58.2)	853 (50.1)	716 (59.9)	944 (58.9)
	High	496 (28.2)	715 (42.0)	274 (22.9)	415 (25.9)

Cronbach’s α = 0.75 (internal consistency for Scale Symptoms).

**Table 2 ijerph-20-06473-t002:** Predicted psychological symptoms, by gender and family affluence scale, in the Tuscany HBSC 2006–2018 surveys.

	Family Affluence Scale	2006 Predicted Score (95% CI)	2010 Predicted Score (95% CI)	2014 Predicted Score (95% CI)	2018 Predicted Score (95% CI)	Difference 2018–2006
Girls						
	Low	7.09 (7.00, 7.17)	6.85 (6.73, 6.97)	6.82 (6.73, 6.91)	7.37 (7.29, 7.47)	0.29 (0.17, 0.41)
	Medium	6.27 (6.23, 6.31)	6.39 (6.35, 6.44)	6.79 (6.75, 6.84)	7.70 (7.66, 7.74)	1.43 (1.37, 1.48)
	High	6.43 (6.37, 6.49)	6.62 (6.57, 6.67)	7.12 (7.05, 7.19)	7.86 (7.79, 7.92)	1.43 (1.34, 1.52)
Boys						
	Low	5.60 (5.51, 5.69)	5.49 (5.38, 5.59)	5.13 (5.03, 5.22)	5.74 (5.65, 5.84)	0.14 (0.01, 0.27)
	Medium	5.20 (5.16, 5.24)	5.22 (5.18, 5.27)	5.03 (4.99, 5.08)	5.91 (5.87, 5.96)	0.71 (0.65, 0.77)
	High	5.29 (5.24, 5.35)	5.35 (5.30, 5.39)	5.75 (5.68, 5.83)	5.60 (5.53, 5.67)	0.31 (0.22, 0.39)

Psychological symptoms (depressed, irritable, nervous, sleep difficulties) every week, or more often, in the previous 6 months.

**Table 3 ijerph-20-06473-t003:** Estimates from the linear regression model of psychological symptom score (range 0–16), HBSC 2006–2018 surveys.

		Regression Coefficient	95% CI	*p*-Value
Gender				
	Girl	Reference		
	Boy	−1.40	−2.07, −0.73	<0.001
Family affluence scale				
	Low	Reference		
	Medium	−0.67	−1.18, −0.17	0.009
	High	−0.47	−1.05, 0.12	0.12
Year of survey				
	2006	Reference		
	2010	0.91	−0.72, 0.91	0.83
	2014	−0.18	−0.86, 0.50	0.60
	2018	0.34	−0.34, 1.03	0.32
Age (in years)		0.39	0.35, 0.43	<0.001
Gender/Family affluence scale				
	Boy/Medium	0.31	−0.45, 1.06	0.42
	Boy/High	0.17	−0.68, 1.02	0.69
Gender/Year				
	Boy/2010	0.14	−1.02, 1.31	0.81
	Boy/2014	−0.26	−1.26, 0.74	0.62
	Boy/2018	−0.27	−1.26, 0.72	0.59
Family affluence scale/Year				
	Medium/2010	0.15	−0.74, 1.03	0.75
	High/2010	0.13	−0.83, 1.08	0.79
	Medium/2014	0.67	−0.10, 1.44	0.09
	High/2014	0.70	−0.20, 1.60	0.13
	Medium/2018	0.98	0.22, 1.74	0.01
	High/2018	0.90	0.01, 1.78	0.05
Gender/Family affluence scale/Year				
	Boy/Medium/2010	−0.23	−1.50, 1.04	0.73
	Boy/High/2010	−0.17	−1.52, 1.17	0.80
	Boy/Medium/2014	−0.46	−1.58, 0.67	0.43
	Boy/High/2014	0.16	−1.14, 1.46	0.81
	Boy/Medium/2018	0.56	−1.56, 0.65	0.42
	Boy/High/2018	−0.75	−2.00, 0.50	0.24

/: this symbol is used as a word separator.

**Table 4 ijerph-20-06473-t004:** Predicted psychological symptoms (depressed, irritable, nervous, sleep difficulties) every week, or more often, in the previous 6 months by gender and family affluence scale, in the Tuscany HBSC 2006–2018 surveys.

	Family Affluence Scale	2006 Predicted Score (95% CI)	2010 Predicted Score (95% CI)	2014 Predicted Score (95% CI)	2018 Predicted Score (95% CI)	Difference 2018–2006
GIRLS						
Depressed	Low	0.66 (0.65, 0.66)	0.64 (0.63, 0.65)	0.61 (0.60, 0.62)	0.58 (0.57, 0.59)	−0.08 (−0.09, −0.06)
	Medium	0.60 (0.59, 0.60)	0.59 (0.58, 0.59)	0.61 (0.61, 0.62)	0.62 (0.62, 0.63)	0.03 (0.02, 0.03)
	High	0.61 (0.60, 0.61)	0.57 (0.56, 0.57)	0.58 (0.57, 0.59)	0.61 (0.60, 0.62)	0.00 (−0.01, 0.01)
GIRLS						
Irritable	Low	0.60 (0.59, 0.61)	0.52 (0.51, 0.54)	0.53 (0.52, 0.54)	0.59 (0.58, 0.60)	−0.01 (−0.02, 0.01)
	Medium	0.53 (0.52, 0.53)	0.52 (0.51, 0.53)	0.55 (0.54, 0.56)	0.64 (0.63, 0.64)	0.11 (0.10, 0.12)
	High	0.54 (0.53, 0.55)	0.56 (0.56, 0.57)	0.56 (0.55, 0.57)	0.65 (0.64, 0.66)	0.11 (0.10, 0.12)
GIRLS						
Nervous	Low	0.64 (0.63, 0.65)	0.59 (0.58, 0.61)	0.61 (0.60, 0.62)	0.65 (0.64, 0.66)	0.02 (0.00, 0.03)
	Medium	0.61 (0.61, 0.62)	0.59 (0.58, 0.59)	0.62 (0.62, 0.63)	0.68 (0.67, 0.68)	0.06 (0.06, 0.07)
	High	0.58 (0.57, 0.59)	0.63 (0.63, 0.64)	0.62 (0.62, 0.63)	0.69 (0.68, 0.69)	0.10 (0.09, 0.11)
GIRLS						
Sleep difficulties	Low	0.29 (0.29, 0.29)	0.31 (0.31, 0.31)	0.31 (0.31, 0.31)	0.42 (0.42, 0.42)	0.13 (0.13, 0.13)
	Medium	0.23 (0.23, 0.23)	0.28 (0.28, 0.28)	0.29 (0.29, 0.29)	0.40 (0.40, 0.40)	0.17 (0.17, 0.17)
	High	0.25 (0.25, 0.25)	0.31 (0.31, 0.31)	0.37 (0.37, 0.37)	0.40 (0.40, 0.40)	0.15 (0.15, 0.15)
BOYS						
Depressed	Low	0.46 (0.45, 0.47)	0.48 (0.47, 0.49)	0.41 (0.40, 0.42)	0.44 (0.43, 0.45)	−0.02 (−0.03, −0.01)
	Medium	0.41 (0.40, 0.41)	0.43 (0.42, 0.43)	0.41 (0.40, 0.41)	0.46 (0.45, 0.46)	0.05 (0.05, 0.06)
	High	0.43 (0.42, 0.44)	0.42 (0.42, 0.43)	0.46 (0.45, 0.47)	0.39 (0.38, 0.40)	−0.04 (−0.05, −0.03)
BOYS						
Irritable	Low	0.43 (0.42, 0.44)	0.39 (0.37, 0.40)	0.43 (0.42, 0.44)	0.43 (0.42, 0.45)	0.01 (−0.01, 0.02)
	Medium	0.46 (0.46, 0.47)	0.43 (0.43, 0.44)	0.41 (0.40, 0.41)	0.49 (0.49, 0.50)	0.03 (0.02, 0.04)
	High	0.44 (0.43, 0.45)	0.45 (0.44, 0.46)	0.46 (0.45, 0.47)	0.49 (0.48, 0.50)	0.05 (0.04, 0.06)
BOYS						
Nervous	Low	0.54 (0.53, 0.55)	0.54 (0.53, 0.56)	0.44 (0.43, 0.45)	0.52 (0.51, 0.53)	0.01 (−0.03, −0.00)
	Medium	0.50 (0.50, 0.51)	0.50 (0.50, 0.51)	0.46 (0.45, 0.46)	0.56 (0.56, 0.57)	0.06 (0.05, 0.07)
	High	0.49 (0.48, 0.50)	0.55 (0.54, 0.56)	0.52 (0.52, 0.53)	0.56 (0.55, 0.57)	0.07 (0.06, 0.08)
BOYS						
Sleep difficulties	Low	0.23 (0.23, 0.23)	0.25 (0.25, 0.25)	0.24 (0.24, 0.24)	0.30 (0.30, 0.30)	0.07 (0.07, 0.07)
	Medium	0.20 (0.20, 0.20)	0.23 (0.23, 0.23)	0.25 (0.25, 0.25)	0.32 (0.32, 0.32)	0.13 (0.13, 0.13)
	High	0.19 (0.19, 0.19)	0.24 (0.24, 0.24)	0.32 (0.32, 0.32)	0.28 (0.28, 0.28)	0.09 (0.09, 0.09)

## Data Availability

The data presented in this study are available in accordance with the Italian HBSC data access policy. Requests, upon reasonable request, should be directed to giacomo.lazzeri@unisi.it, Tuscan coordinator for HBSC program.

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
