# Peer review of "Trend of Correlations between Psychological Symptoms and Socioeconomic Inequalities among Italian Adolescents: Cross-Sectional Study from 2006 to 2018 in Tuscany Region"

_ijerph, 2023, doi:10.3390/ijerph20156473_

Round 1
Reviewer 1 Report
The aim of this study is to investigate the association between psychological disorders and socioeconomic status in adolescents in Tuscan schools, aged 11, 13 and 15 years. I thank the authors for tackling such a relevant and current topic.
Some general considerations:
- I consider it important to broaden the introduction and discussion by referring to known variables of the phenomenon and related to vertical and horizontal social interaction, in terms of quantity and especially quality (primary emotions, satisfaction, happiness, self-esteem, etc.). These variables have a very important influence on the estimation of the observed phenomenon;
- Studies of a sociological and psychosocial (not anthropological) nature relating to the well-being of the population group observed should be mentioned
- The study concentrates the analysis up to 2018 (today the situation is different as the negative psychological and psychosocial effects produced by the spread of Covid-19 are known)
Specific considerations:
- it is necessary to insert the research hypotheses at the end of the introductory paragraph;
- it is necessary to specify which specific role was played by the various professional figures involved in the survey (principals, teachers and others);
- it is necessary to specify the specific methods of application of the standardized scientific methods used for the purposes of the analysis;
- it is necessary to specify that the total sample (12,550 participants) concerns the sum of four different samples since it is a transversal study;
- it is necessary to clarify the nature of the scale used to measure psychological symptoms, also with specific bibliographic references;
- it is necessary to broaden the conclusions with respect to the limits of the study (for example the limited variables analysed). It would also be useful to hypothesize, in the light of the results, specific intervention proposals concerning the youth universe also in the school and family sphere with bibliographic references to the social and psychological effects of the pandemic: the spread of COVID-19, in fact, as known in the scientific literature , have also worsened the situation in terms of youth well-being due to the increase in virtual social interaction, and consequently due to the accentuation of some phenomena such as cyberbullying and online violence
REQUESTS:
- Were the 4 samples built the same way?
- How were the research units (students) selected?
- How likely were all the students to access the survey?
- How to administer the questionnaire?
- In whose presence was the questionnaire completed in schools?
Author Response
I have attached the file

Reviewer 2 Report
The study under review was performed by Italian medical and public health experts. It is not known whether psychological test specialists took part.
The research field is highly complex and interrelated (a recent US topic page of science.gov on family affluence scale lists 1,500 entries!). The scientific format and empirical work of the study looks good and is self-critical in the discussion. Re-checks of the FAS scale (e.g. Torsheim 2011, Hartley 2015, see below) suggest that it has few valid items and remains a weak point when examining adolescent socioeconomic inequalitities. But this is not a fault of the authors under review.
It remains unclear for the reviewer whether the 27 quote references constitute a valid and balanced research sample. Useful additional references would be:
Levin et al. 2011 Social Indicators Research (DOI 10.1007/s11205-010-9747-8) who reviewed N=53,352 from 35 countries and used a 4 item FAS scale. Socioeconomic inequalities were greater in poor countries and under unequal income distributions.
Torsheim et al. 2011 Child Indicators Research (DOI 10.1007/s12187-015-9339-x) covered N=7,120 in eight countries and found only six valid items for the FAS.
Hartley et al. 2015 Child Indicators Research (DOI: 10.1007/s12187-015-9325-3) mentioned Scottish results and considered the FAS needed to be refined.
Voracova et al. 2016 IJERPH (doi:10.3390/ijerph13101034) studied eating habits of N=5012+5819 11-15 years old and found better health behavior for higher FAS scores (4 items).
A minor flaw is that the interaction calculations using regressions with "Year" suggest a repetition of measurements that did not take place, actually - all test persons were only examined once and form different independent samples. The question remains what the interaction calculations with the respective survey years of the sub-samples 2006, 2010, 2014 and 2018 are supposed to say except that they do not differ significantly.
For future studies or a repetition, a statistical suggestion: It should be considered why the evaluation in the study under review was done by linear and ordinal regressions instead of testing the underlying difference hypotheses (main effects and interactions) with (co-)variance analyzes with subsequently performed (alpha-error-corrected) post-hoc tests. The measurement tools used in the reviewed study could be used for a more accurate report including the description of the individual item formulations, any test performance statistics additions (e.g. part-whole corrected power levels, not just Cronbach's alpha) in order to better assess the statements derived for the investigation hypotheses (also with regard to validity).
Author Response
Dear Reviewer
Thank you for the review work and suggestions provided, very interesting and pointers to improve this and the work we are completing.
We have included the reported references so that the work is more balanced and valuable.
Q: A minor flaw is that the interaction calculations using regressions with "Year" suggest a repetition of measurements that did not take place, actually - all test persons were only examined once and form different independent samples. The question remains what the interaction calculations with the respective survey years of the sub-samples 2006, 2010, 2014 and 2018 are supposed to say except that they do not differ significantly.
R: I understand that the samples can be considered independent, since for each subject only one measurement is done per year (and the subjects are different between the various years) and in effect it is a Cross-Sectional study, without repeated measurements. This could be considered as a limitation of the study.
The two- or three-way interaction was inserted in the linear regression model to verify the presence of a multiplicative effect between the independent variables, therefore, to highlight statistically significant differences between the various factors.
I agree that this type of analysis could be done without using regression models but analysis of (co)-variance and post-hoc tests (used correctly in case of independent samples), but in this case, the approach used was followed by the main objective of the article, namely, to measure the trend of the correlation between psychological symptoms and socioeconomic inequalities between 2006 and 2018, then, with the linear regression model, to show the trend for psychological symptoms over the various years adjusted for age, while with logistic regression models, measure the probability of developing each symptom once or more a week, in relation to gender and socioeconomic status in the years, also highlighting the difference between the two extreme years of observation (2006 and 2018).
Round 2
Reviewer 1 Report
At the end of the introductory paragraph the objective has been confirmed but the research hypotheses, which have been subjected to statistical analysis, are not defined
Author Response
Dear reviewer, thank you for your comment, I have carried out the request. I hope to have interpreted the request correctly.
